# Essential role for SUN5 in anchoring sperm head to the tail

**Yongliang Shang[1,2†], Fuxi Zhu[3,4†], Lina Wang[1,2], Ying-Chun Ouyang[1], Ming-Zhe Dong[1], Chao Liu[1], Haichao Zhao[1,2], Xiuhong Cui[1], Dongyuan Ma[5], Zhiguo Zhang[3,4], Xiaoyu Yang[6], Yueshuai Guo[7], Feng Liu[2,5], Li Yuan[8], Fei Gao[1,2*], Xuejiang Guo[7*], Qing-Yuan Sun[1,2*], Yunxia Cao[3,4*], Wei Li[1,2*]**

[1]State Key Laboratory of Stem Cell and Reproductive Biology, Institute of Zoology, Chinese Academy of Sciences, Beijing, China; [2]University of Chinese Academy of Sciences, Beijing, China; [3]Reproductive Medicine Center, Department of Obstetrics and Gynecology, The First Affiliated Hospital of Anhui Medical University, Hefei, China; [4]Institute of Reproductive Genetics, Anhui Medical University, Hefei, China; [5]State Key Laboratory of Membrane Biology, Institute of Zoology, Chinese Academy of Sciences, Beijing, China; [6]Center of Clinical Reproductive Medicine, First Affiliated Hospital, Nanjing Medical University, Nanjing, China; [7]State Key Laboratory of Reproductive Medicine, Collaborative Innovation Center of Genetics and Development, Department of Histology and Embryology, Nanjing Medical University, Nanjing, China; [8]Savaid School of Medicine, University of Chinese Academy of Sciences, Beijing, China

**\*For correspondence:**
gaof@ioz.ac.cn (FG);
guo_xuejiang@njmu.edu.cn (XG);
sunqy@ioz.ac.cn (Q-YS);
caoyunxia6@126.com (YC);
leways@ioz.ac.cn (WL)

[†]These authors contributed equally to this work

**Abstract** SUN (Sad1 and UNC84 domain containing)-domain proteins are reported to reside on the nuclear membrane playing distinct roles in nuclear dynamics. SUN5 is a new member of the SUN family, with little knowledge regarding its function. Here, we generated $Sun5^{-/-}$ mice and found that male mice were infertile. Most $Sun5$-null spermatozoa displayed a globozoospermia-like phenotype but they were actually acephalic spermatozoa. Additional studies revealed that SUN5 was located in the neck of the spermatozoa, anchoring sperm head to the tail, and without functional SUN5 the sperm head to tail coupling apparatus was detached from nucleus during spermatid elongation. Finally, we found that healthy heterozygous offspring could be obtained via intracytoplasmic injection of $Sun5$-mutated sperm heads for both male mice and patients. Our studies reveal the essential role of SUN5 in anchoring sperm head to the tail and provide a promising way to treat this kind of acephalic spermatozoa-associated male infertility.
DOI: https://doi.org/10.7554/eLife.28199.001

## Introduction

The SUN domain proteins were named for their shared homologous sequences with Sad1 and UNC-84 (*Malone et al., 1999*); the former is an essential component of the spindle body in fission yeast (*Hagan and Yanagida, 1995*), and the latter is a nuclear membrane protein that mediates nuclear migration and positioning in the nematode *Caenorhabditis elegans* (*Malone et al., 1999*). In mammals, at least five SUN domain proteins have been reported, and three of these genes in mice have been named *Sun1*, *Sun2*, and *Sun3* (*Malone et al., 1999*; *Crisp et al., 2006*). Two other SUN domain proteins were originally named rat sperm-associated antigen 4 (*SPAG4*) (*Tarnasky et al., 1998*; *Shao et al., 1999*) and SPAG4-like (*SPAG4L*) (*Jiang et al., 2011*), by sequentially, and they are coded by *Sun4* and *Sun5*.

The SUN domain proteins possess transmembrane domains in their N-terminus and a conserved SUN domain in their C-terminus (*Mans et al., 2004*). The transmembrane domain has been proven to integrate the SUN proteins into the inner membrane of the nuclear envelope, with their N-terminus facing the nucleoplasm (*Hodzic et al., 2004*). It is thought that the nucleoplasmic N-terminus of SUN proteins could interact with nuclear lamin proteins and fasten the linkage between the nuclear envelope and the nucleoplasm or chromatins (*Crisp et al., 2006*; *Haque et al., 2006*; *Wang et al., 2006*). The SUN domain in the C-terminus of SUN proteins has been reported to interact with various components in the outer nuclear membrane, mainly cytoskeleton-associated proteins (*Apel et al., 2000*) that contain a conserved KASH (Klarsicht, ANC-1 and Syne homology) domain (*Starr and Han, 2002*). In this way, the SUN family builds a bridge between the nucleoskeleton and the cytoskeleton, forming the so-called LINC complexes (linker of nucleoskeleton and cytoskeleton) (*Crisp et al., 2006*; *Stewart-Hutchinson et al., 2008*) and mediating nuclear dynamics during mitosis or meiosis (*Stewart et al., 2007*; *Fridkin et al., 2004*; *Tapley and Starr, 2013*; *Kracklauer et al., 2013*).

SUN1 and SUN2 are two well-studied SUN proteins that are broadly expressed in both mitotic and meiotic cells (*Padmakumar et al., 2005*). SUN1 is linked to F-actin filaments across the outer nuclear membrane-residing Nesprine1/2 to stabilize the nuclear anchorage and maintain nuclear envelope integrity (*Yu et al., 2011*). SUN1 is also linked to microtubules via KASH5 on the outer nuclear membrane, mediating telomere attachment to the nuclear envelope during meiosis (*Morimoto et al., 2012*; *Ding et al., 2007*; *Penkner et al., 2009*). Recent studies have found that SUN1 could mediate mammalian mRNA export (*Li and Noegel, 2015*). SUN2 shares similar interactors with SUN1 and performs related functions in nuclear envelope integrity and telomere attachment. In addition, it is thought that SUN1 and SUN2 play several redundant roles in this anchoring mechanism (*Lei et al., 2009*).

SUN3, SUN4 and SUN5, which are shorter than SUN1 and SUN2, are expressed restrictively in testes (*Hiraoka and Dernburg, 2009*). SUN3 has been reported to localize to the manchette in elongating spermatids, which is distinct from the classical nuclear membrane localization of the SUN family proteins, and its expression begins at postnatal D15. It is associated with Nesprine1, facilitating sperm head shaping during spermiogenesis (*Göb et al., 2010*). SUN4, which is also distributed in the manchette, can interact with SUN3, indicating the association of their localization and physiological functions. Male mice deficient in SUN4 are infertile due to globozoospermia, and SUN4 can bind to ODF1 (outer dense fiber protein 1), a sperm flagellum protein, suggesting that it might function in either nuclear remodeling (*Calvi et al., 2015*) or sperm integrity (*Shao et al., 1999*) during spermiogenesis.

The story of SUN5 is much more complicated. SUN5 was first found to be localized on the spermatid nuclear membrane facing the acrosome, and it was predicted to participate in acrosome biogenesis or to attach the acrosome to the nuclear envelope (*Frohnert et al., 2011*). However, a recent study in *Dpy19l2* knockout mice found that neither the expression nor the localization of SUN5 was altered, suggesting that it might not be associated with the acrosome; instead, they found that SUN5 localizes to the sperm head-tail junction (*Yassine et al., 2015*). Whether SUN5 is involved in acrosome biogenesis, head-tail integration or nuclear dynamics similar to the functional role of SUN1/2 remains elusive, due to the lack of animal models.

To address the above question, we generated a *Sun5* knockout mouse model via the CRISPR-Cas9 system to study its physiological functions during spermatogenesis. We found that $Sun5^{-/-}$ females were fertile, but the $Sun5^{-/-}$ male mice were sterile. Normal spermatozoa were not found in the epididymis of $Sun5^{-/-}$ male mice, as most of them were round-headed like spermatozoa; a few normal sperm heads could be found, but they were all separated from the sperm flagellum. Further studies uncovered that the so-called round-headed like spermatozoon from $Sun5^{-/-}$ male mice does not contain chromatin or acrosome; instead, it was filled with unremoved cytoplasm and misarranged mitochondria. Therefore, we proposed to name this phenotype of spermatozoa as pseudo-globozoospermia. Ultrastructural studies of spermatogenesis in $Sun5^{-/-}$ male mice revealed that the sperm head-tail coupling apparatus could be successfully assembled during the early stage of spermiogenesis, but without functional SUN5, the coupling apparatus together with the basal plate disassociated from the implantation fossa during the elongation of the spermatids. Most importantly, we found that healthy offspring could be obtained from $Sun5^{-/-}$ sterile male mice and patients by microinjection of the tailless sperm head into the oocyte. Our investigations not only settled the

dispute about the physiological function of SUN5 but also provided a successful therapeutic strategy for *SUN5*-deficient patients. Our studies suggest that the sperm head needs to be carefully evaluated before ICSI for teratozoospermia patients to avoid this type of pseudo-globozoospermia.

## Results

### The generation of *Sun5* knockout mice

To investigate the expression pattern of *Sun5*, we tested the mRNA level of *Sun5* in various tissues from adult mice, finding that *Sun5* was strictly expressed in testes (*Figure 1—figure supplement 1A*). Further examination of testes from different-aged mice found that SUN5 expression began in the 3-week-old mouse testes, suggesting that it might participate in certain processes of spermiogenesis (*Figure 1—figure supplement 1B*).

Because *Sun5* is restricted to the testis, we applied the CRISPR-Cas9 system to achieve *Sun5* knockout. One Cas9-targeting sequence was found in exon 10 of the *Sun5* locus, which encodes the conserved SUN domain together with exon 11 and exon 12 (*Figure 1—figure supplement 1C*). One *Sac*II site was found near the PAM (protospacer-adjacent motif) sequence, which is the only *Sac*II site in approximately 500 bp both upstream and downstream of the Cas9-targeting sequence. For genotyping of the mutated mice, a 538 bp fragment harboring the PAM sequence and *Sac*II site was amplified from the genome and then digested using the *Sac*II enzyme. The WT genome was cut into two fragments, a 303 bp and a 235 bp fragment, while the mutated genome remained undigested.

This strategy yielded *Sun5* mutated mice with relatively high efficiency (*Figure 1—figure supplement 1D*), five founders were identified among eight pups. After cloning and subsequent sequencing, the mutated sequences were identified: four of the five founders were biallelic mutants, while one was a heterozygous-mutated mouse (*Figure 1—figure supplement 1E*). All of the *Sun5* mutated mice were viable and developed normally, but all of the male biallelic mutants failed to produce offspring, while the female biallelic mutants and heterozygous-mutated males were fertile (*Figure 1—figure supplement 1F*). To establish the *Sun5* mutant strain, we chose the fertile heterozygous-mutated male mouse ($Sun5^{+/-13bp}$) to keep the mutated allele and breed *Sun5* knockout mice. Therefore, all of the $Sun5^{-/-}$ mice used in this study were $Sun5^{-13bp/-13bp}$ mice.

### Defects of spermiogenesis in $Sun5^{-/-}$ mice

We found that the SUN5 protein was completely depleted in the $Sun5^{-/-}$ strain but not in the WT and $Sun5^{+/-}$ males (*Figure 1A*), and there were no significant differences in the viability and testis weight among the three strains (*Figure 1B* and *Figure 1—figure supplement 2A–B*). Next, we performed a systematic fertility test and found that the $Sun5^{-/-}$ male mice were sterile, in contrast with the $Sun5^{+/-}$ and WT mice (*Figure 1C–D*). Histologically, the testicular component of $Sun5^{-/-}$ mice was similar to that of $Sun5^{+/-}$ and WT mice (*Figure 1E*), while the contents in the epididymis of $Sun5^{-/-}$ mice were different from that of $Sun5^{+/-}$ and WT mice. The $Sun5^{+/-}$ and WT mouse epididymides were filled with blue-stained sperm heads as well as red-stained flagella and droplets, but in the $Sun5^{-/-}$ mouse epididymis, blue-stained sperm heads could hardly be found. Instead, red round-headed spermatozoa were observed (*Figure 1F*), suggesting that *Sun5* knockout might affect sperm head formation. Further analysis of the spermatozoa from the caudal epididymis did not reveal an obvious difference in the sperm concentration among the $Sun5^{-/-}$, $Sun5^{+/-}$ and WT mice (*Figure 1G*), while the percentage of motile spermatozoa was significantly decreased in the $Sun5^{-/-}$ mice (*Figure 1H*), and most of the motile *Sun5*-null spermatozoa belonged to medium- or slow-moving groups according to the CASA (Computer-aided sperm analysis) (*Figure 1—figure supplement 2C*). Further examinations did not find healthy spermatozoa in the epididymis of $Sun5^{-/-}$ mice, most of them were round-headed and only a few normal sperm heads were observed, but they were all separated from the sperm flagella, which were rarely observed in $Sun5^{+/-}$ and WT mice (*Figure 1I–J*). The proportion of the round-headed spermatozoa and tailless heads were shown in *Figure 1J*. We then evaluated the three sterile founders and observed similar staining patterns in the testis and epididymis of these male mice (*Figure 1—figure supplement 3A*). Additionally, the spermatozoa in these mice were also round-headed (*Figure 1—figure supplement 3B*), indicating that the sterile phenotype of all *Sun5*-mutated male mice resulted from the same reason, thus ensuring that $Sun5^{-13bp/-13bp}$ mice could be used in the following mechanistic studies. As round-

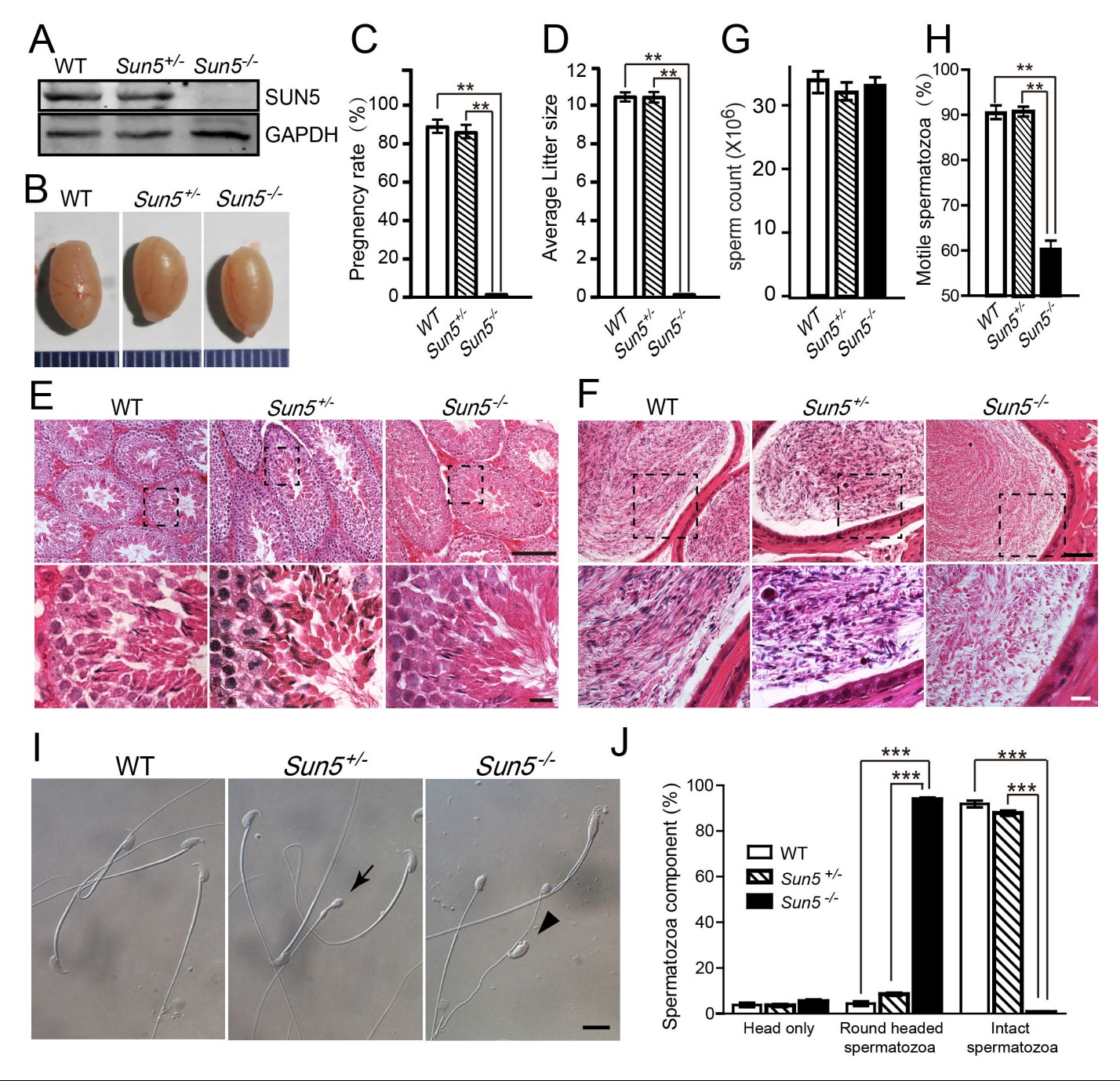

**Figure 1.** Ablation of SUN5 leads to male infertility and sperm malformation. (A) Immunoblotting of SUN5 in WT, $Sun5^{+/-}$ and $Sun5^{-/-}$ testes. (B) The size of the testes was not altered in the $Sun5^{+/-}$ and $Sun5^{-/-}$ mice. (C) The pregnancy rate of WT (92.46 ± 3.39%), $Sun5^{+/-}$ (88.33 ± 3.73%) and $Sun5^{-/-}$ (0) male mice (n = 6). $p_{(WT\ VS\ Sun5-/-)} = 1.24 \times 10^{-6}$, $p_{(Sun5+/-\ VS\ Sun5-/-)} = 1.37 \times 10^{-7}$, $p_{(WT\ VS\ Sun5+/-)} = 0.29$. (D) The average litter size of WT (10.65 ± 0.21), $Sun5^{+/-}$ (10.27 ± 0.38) and $Sun5^{-/-}$ (0) male mice (n = 6). $p_{(WT\ VS\ Sun5-/-)} = 9.85 \times 10^{-27}$, $p_{(Sun5+/-\ VS\ Sun5-/-)} = 5.47 \times 10^{-25}$, $p_{(WT\ VS\ Sun5+/-)} = 0.39$. (E) HE (hematoxylin-eosin) staining of testes from WT, $Sun5^{+/-}$ and $Sun5^{-/-}$ mice, seminiferous tubules shown in the figures were at stage IV-VI. Scale bar: upper panel, 100 μm; lower panel, 10 μm. (F) HE staining of the caudal epididymis from WT, $Sun5^{+/-}$ and $Sun5^{-/-}$ mice. Scale bar: upper panel, 50 μm; lower panel, 10 μm. (G) The sperm concentration of WT (33.92 ± 1.71 × $10^6$), $Sun5^{+/-}$ (31.29 ± 0.93 × $10^6$) and $Sun5^{-/-}$ (33.03 ± 1.67 × $10^6$) male mice (n = 5). $p_{(WT\ VS\ Sun5^{-/-})} = 0.65$, $p_{(Sun5^{+/-}\ VS\ Sun5^{-/-})} = 0.41$, $p_{(WT\ VS\ Sun5^{+/-})} = 0.17$ (H) The percentage of motile spermatozoa in WT (90.20 ± 0.63%), $Sun5^{+/-}$ (90.40 ± 0.14%) and $Sun5^{-/-}$ (60.2 ± 1.98%) male mice (n = 5). $p_{(WT\ VS\ Sun5-/-)} = 0.0001$, $p_{(Sun5+/-\ VS\ Sun5-/-)} = 0.0005$, $p_{(WT\ VS\ Sun5+/-)} = 0.92$. (n = 5) (I) Caudal epididymal spermatozoa of WT, $Sun5^{+/-}$ and $Sun5^{-/-}$ mice. The arrow indicates the round-headed spermatozoon in $Sun5^{+/-}$ mice, and the arrowhead indicates the tailless head spermatozoon in $Sun5^{-/-}$ mice. Scale bar: 10 μm. (J) The percentage of different spermatozoon components in WT, $Sun5^{+/-}$ and $Sun5^{-/-}$ caudal epididymides (n = 5). The first group of columns show the percentage of isolated sperm heads in WT (3.78 ± 0.90%), $Sun5^{+/-}$ (3.60

*Figure 1 continued on next page*

*Figure 1 continued*

± 0.62%) and $Sun5^{-/-}$ (5.44 ± 0.79%) mice, $p_{(WT\ VS\ Sun5-/-)}$= 0.32, $p_{(Sun5+/-\ VS\ Sun5-/-)}$= 0.07; The 2nd group of columns show the percentage of round headed spermatozoon in WT (4.38 ± 0.96%), $Sun5^{+/-}$ (8.56 ± 0.06%) and $Sun5^{-/-}$ (93.86 ± 0.79%) mice, $p_{(WT\ VS\ Sun5-/-)}$= 5.82 × $10^{-7}$, $p_{(Sun5+/-\ VS\ Sun5-/-)}$= 2.73 × $10^{-7}$; The 3rd group of columns show the percentage of intact spermatozoon in WT (92.44 ± 1.63%), $Sun5^{+/-}$ (87.84 ± 1.13%) and $Sun5^{-/-}$ (0.7 ± 0.28%) mice, $p_{(WT\ VS\ Sun5-/-)}$= 1.01 × $10^{-6}$, $p_{(Sun5+/-\ VS\ Sun5-/-)}$= 7.63 × $10^{-8}$. Data represent mean ±SEM.
DOI: https://doi.org/10.7554/eLife.28199.002

The following source data and figure supplements are available for figure 1:

**Source data 1.** Source data for mouse fertility, sperm concentration, sperm motility and spermatozoa components in epididymis.
DOI: https://doi.org/10.7554/eLife.28199.006
**Figure supplement 1.** The generation of *Sun5* knockout mice.
DOI: https://doi.org/10.7554/eLife.28199.003
**Figure supplement 2.** *Sun5* knockout does not affect growth and testis development, but influences sperm motility.
DOI: https://doi.org/10.7554/eLife.28199.004
**Figure supplement 3.** All of the biallelic *Sun5* mutated male mice exhibit similar defects in spermiogenesis.
DOI: https://doi.org/10.7554/eLife.28199.005

headed spermatozoa or globozoospermia usually result from an acrosome biogenesis defect or a complete loss of acrosome, these results appear to support a physiological function for SUN5 in acrosome biogenesis.

## The 'round-headed' *Sun5*-null spermatozoa are actually headless sperm flagella

To confirm whether the round-headed spermatozoa found in $Sun5^{-/-}$ mice are typical globozoospermatozoa, we examined the development of the acrosome, the key organelle of the spermatozoa. Acrosome biogenesis can be divided into four developmental phases according to its biogenesis: Golgi phase, cap phase, acrosome phase and maturation phase (*Wang et al., 2014*). To our surprise, we did not find any defect in acrosome biogenesis in the testis of $Sun5^{-/-}$ mice, and all of the four typical developmental phases could be found in both WT and $Sun5^{-/-}$ mice (*Figure 2— figure supplement 1A*). These results suggest that the round-headed spermatozoa in $Sun5^{-/-}$ male mice are not resulted from abnormal acrosome biogenesis, and SUN5 might not participate in this process.

To further test whether *Sun5* knockout has any impact on the acrosome, single-sperm immunofluorescence was performed using the acrosome-specific marker sp56, and DAPI was co-stained to indicate the nucleus. As mentioned above, the *Sun5*-null spermatozoa contain both round-headed spermatozoa and tailless heads (their proportions were displayed in *Figure 1J*). To our surprise, the round-headed *Sun5*-null spermatozoa were negative for both sp56 and DAPI staining, indicating the absence of not only the acrosome but also the nucleus. After careful examination, we found some separated but morphologically normal sperm heads, most of them had an intact acrosome and a nucleus, and only a small amount of them had defective acrosomes (*Figure 2A*, *Figure 2—figure supplement 1B*). We then measured the width and length of WT and *Sun5*-null sperm heads (*Fisher et al., 2016*), finding that the *Sun5*-null sperm heads were wider and shorter than those of the WT ones (*Figure 2—figure supplement 1C–E*). This promoted us to investigate what exactly happened inside the so-called round-headed *Sun5*-null spermatozoa, so we performed transmission electron microscopy (TEM) analysis of the epididymis from both WT and $Sun5^{-/-}$ mice. As shown in *Figure 2B*, the WT sperm head was well shaped and filled with chromatin, and the sperm head and tail were tightly connected to each other, but this was never observed in the *Sun5*-null sperm head. The so-called round-headed *Sun5*-null sperm actually contained a residual droplet of cytoplasm at the top of the flagellum with misarranged mitochondria inside. In addition, the axoneme of *Sun5*-null sperm was also impaired (*Figure 2B*). Therefore, the *Sun5*-null spermatozoa are actually acephalic spermatozoa or sperm tails only. So we analyzed the ratio of sperm with head versus tails only in the caput, corpus and cauda epididymis, finding that the ratios were all very low in all three parts of $Sun5^{-/-}$ epididymis (*Figure 2—figure supplement 1F*). All these results suggest that the sperm heads have been detached from their tails before they enter into the epididymis, and the acephalic tails in Sun5 mutated mice might be the main reason for their infertility.

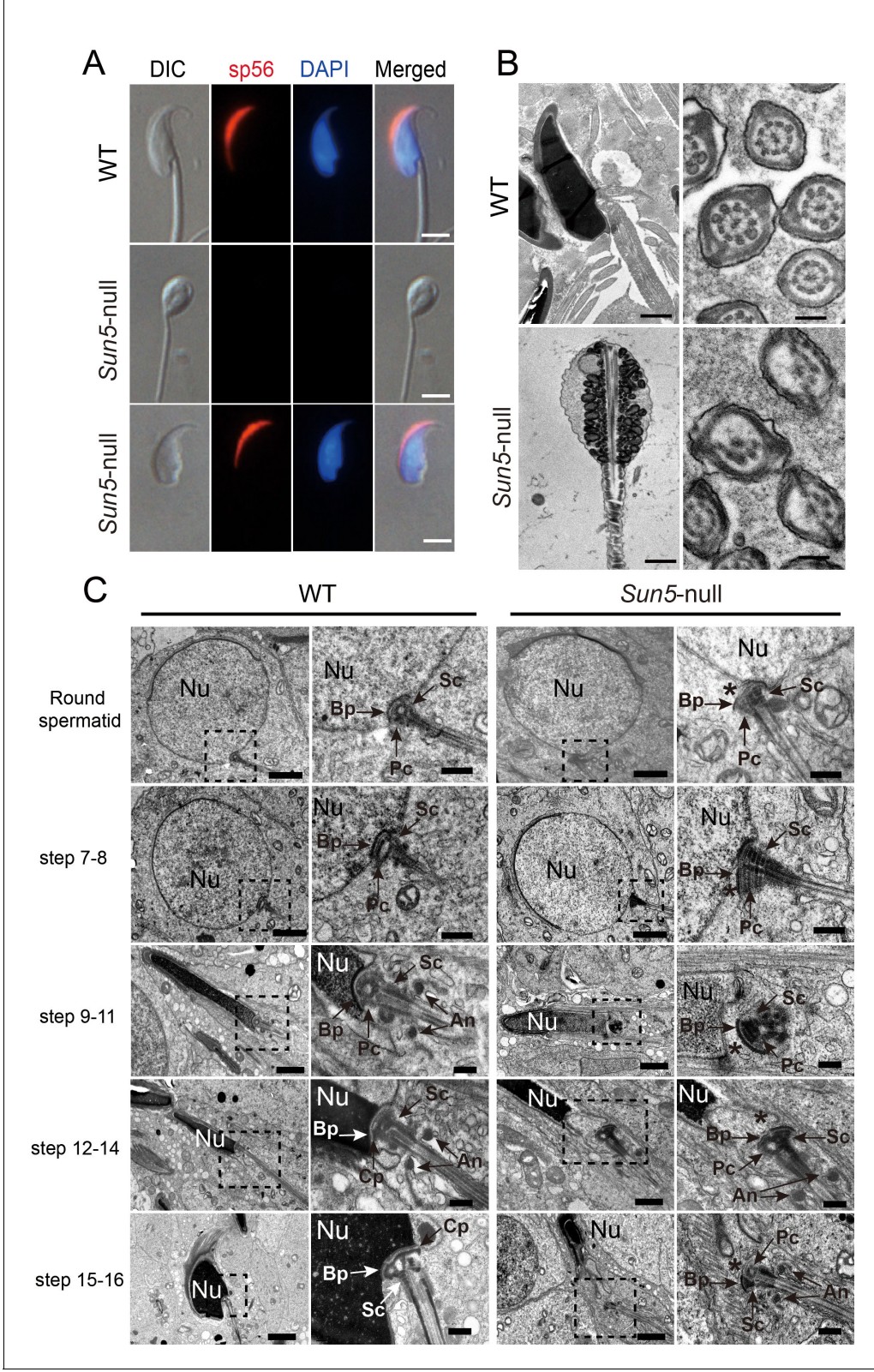

**Figure 2.** The absence of SUN5 has no effect on acrosome biogenesis but disrupts the development of the coupling apparatus between sperm head and tail. (**A**) IF (immunofluorescence) staining of sp56 in WT and *Sun5*-null spermatozoon. The *Sun5*-null spermatozoa contains both round-headed spermatozoa and tailless heads (lower two panels). The proportion of these two types of spermatozoa were displayed in *Figure 1J*. Note that the round-headed *Sun5*-null spermatozoa do not contain nuclei and acrosomes, but the tailless *Sun5*-null sperm heads have nuclei and acrosomes. Scale

*Figure 2 continued on next page*

*Figure 2 continued*

bar: 5 µm. (**B**) Ultrastructure of WT and *Sun5*$^{-/-}$ caudal epididymides showing that the *Sun5*-null spermatozoon was filled with cytoplasm and misarranged mitochondria. Note that the axoneme of *Sun5*-null spermatozoon was also disrupted. Scale bar: left panel, 1 µm; right panel, 200 nm. (**C**) TEM analyses of the stepwise development of the coupling apparatus in WT and *Sun5*-null spermatozoa. In the round spermatid stage, the coupling apparatus can be assembled in both WT and *Sun5*-null spermatid, but the coupling apparatus could not be tightly attached to the nuclear envelope in *Sun5*-null spermatids. The asterisk indicates the gap between the nuclear (Nu) envelope and the basal plate (Bp). In the following developmental stages, the coupling apparatus was well-fixed on the nuclear envelope in WT spermatids, ensuring healthy spermatid differentiation. While in *Sun5*-null spermatids, the basal plate (Bp)-capitulum (Cp)-segmented column (Sc) together with the centriole (Pc) was detached from the nuclear envelope during spermatid elongation. An, annulus. Scale bar: the 1st and 3rd panel, 2 µm, 2nd and 4th panel, 0.5 µm.

DOI: https://doi.org/10.7554/eLife.28199.007

The following figure supplements are available for figure 2:

**Figure supplement 1.** Acrosome biogenesis and epididymal spermatozoa in *Sun5*$^{-/-}$ testes.

DOI: https://doi.org/10.7554/eLife.28199.008

**Figure supplement 2.** Spermiation defects in *Sun5*$^{-/-}$ mice.

DOI: https://doi.org/10.7554/eLife.28199.009

## SUN5 is responsible for the attachment of the coupling apparatus to the sperm nuclear envelope

To determine when the sperm head and tail break apart and why they are separated in *Sun5*$^{-/-}$ mice, we examined spermiogenesis stage by stage using Periodic Acid-Schiff (PAS) staining. According to the component of the spermatids the seminiferous tubules could be divided into 12 stages (I-XII) (***Hess and Renato de Franca, 2008***). No obvious defects were found at any stage of spermiogenesis, and all of the components in the WT testes could be found in the *Sun5*$^{-/-}$ testes (***Figure 2—figure supplement 2A***). However, with careful examination of stage VII-VIII tubules, we observed a difference between WT and *Sun5*$^{-/-}$ testes. In the WT testis, the well-shaped spermatozoa had migrated to the edge of the seminiferous epithelium with their head and acrosomic system oriented toward the basement membrane (***Figure 2—figure supplement 2B***, *Top*). In *Sun5*$^{-/-}$ testes, it was quite different; although the shape of sperm head was normal, they were not oriented toward the basement membrane, as most of their heads were oriented toward the lumen of the seminiferous tubules (***Figure 2—figure supplement 2B***, *Bottom*); peanut agglutinin (PNA, staining acrosome specifically) staining of testis sections also confirmed the mis-orientation of sperm heads (***Figure 2—figure supplement 2C***). Stage VII-VIII is the so-called spermiation phase when mature spermatozoa are ready to be released. This indicates that the *Sun5*-null sperm head and tail might break apart during spermiation so that the separated sperm head cannot align itself in the right orientation.

These observations allowed us to determine what occurred before the sperm release in *Sun5*$^{-/-}$ testes. For the differentiation of haploid spermatids could be divided into 16 steps, and each step could be recognized via TEM (***Shang et al., 2016***). We then investigated the differentiation of spermatids and the assembly of the head-tail coupling apparatus step by step via TEM. In round spermatids, the head-tail coupling apparatus in both WT and *Sun5*$^{-/-}$ testes was fully developed and consisted of a well-assembled segmented column that united the centrioles at the anterior and formed the capitulum. In WT spermatid, the well-assembled segmented column together with the capitulum and basal plate was tightly attached to the nuclear envelope in the implantation fossa (***Figure 2C***, *Top*), whereas in the *Sun5*-null spermatid, although the segmented column with the capitulum and basal plate was accurately assembled, it was only partially connected with the nuclear envelope; a large part of the coupling apparatus was missing. With the elongation of the spermatid, the WT spermatid coupling apparatus together with the flagellum was always tightly attached to the nuclear envelope until the final step of spermatid differentiation, producing structurally normal spermatozoon. In *Sun5*$^{-/-}$ testes, the elongation of the spermatid destroyed the unstable interaction between the nuclear envelope and the coupling apparatus, resulting in the separation of the basal plate-capitulum-segmented column complex from the nuclear envelope. Therefore, in the *Sun5*$^{-/-}$ testes, the decapitated flagella are released while the sperm heads remain in the seminiferous epithelium. These results indicate that SUN5 is responsible for the tight attachment of the coupling apparatus to the sperm nuclear envelope.

## SUN5 is localized at the sperm head-tail coupling apparatus

All of the observations above led us to rethink the exact function of SUN5 during spermiogenesis. In the mouse testis, we found that in mature spermatozoon, SUN5 was predominantly located in the coupling apparatus between the sperm head and tail, thus supporting its function in sperm head and tail integrity (*Figure 3A*). Using a testis smear, we found that during spermiogenesis, SUN5 was first expressed in the nuclear envelope and later migrated to the coupling apparatus of the sperm during sperm head elongation and differentiation (*Figure 3B*). In mature spermatozoa, SUN5 was localized to the coupling apparatus of the sperm head and tail in the implantation fossa (*Figure 3C*). These results suggest that the function of SUN5 is to connect the sperm head to the tail, thus integrating them into a spermatozoon. Additionally, the results suggest that SUN5 is not associated with acrosome biogenesis or nuclear remodeling.

Protein sequence alignment has found that SUN5 is evolutionarily conserved in mammals, and the phylogenic tree shows that SUN5 has sequence identity over 90% in most mammals such as mouse, rat, human, bovine and sheep (*Figure 3D–E*). We then collected spermatozoa from various mammals including rat, human, bovine and sheep and performed single-sperm immunofluorescence analysis using anti-SUN5 antibody. As expected, the SUN5 antibody could recognize all of the homologues of the SUN5 protein, and all of the proteins localized to the neck of the mammalian spermatozoa (*Figure 3F*). These results suggest that the function of SUN5 protein might be evolutionarily conserved in mammals. We summarized the stepwise development of WT and *Sun5*-null spermatids and indicated the potential function of SUN5 during spermiogenesis in *Figure 3G*.

## Overcoming SUN5 defect by ICSI

Consistent with our analysis, our recent survey of some acephalic spermatozoa patients found a series of biallelic mutations in the *SUN5* gene that affected 47.06% of the investigated patients (*Zhu et al., 2016*). And the majority of the *SUN5*-mutated spermatozoa were actually sperm tails with low motility (*Figure 2—figure supplement 1F*, *Figure 4—source data 1*), so an effective method to produce a healthy baby for these infertile patients and their families is urgently needed. Since the $Sun5^{-/-}$ mice are quite similar to the *SUN5*-mutated patients in terms of the phenotype of their spermatozoa, this mouse model provides a good platform upon which to find the proper therapeutic strategy for those patients. Given that *Sun5*-null spermatozoa are actually pseudo-globozoospermia that do not contain chromatin in the round head, the regular ICSI protocol cannot be applied to these mice or patients. The traditional ICSI method favors the selection of relatively intact spermatozoa, then, after sonication, the tailless heads are injected into the oocyte, which is not applicable to the *Sun5* mutants.

To achieve a successful pregnancy, we selected the tailless heads of *Sun5*-null spermatozoa rather than the pseudo-globozoospermatozoa and injected them into WT oocytes; this strategy resulted in healthy offspring from $Sun5^{-/-}$ mice (*Figure 4A–B*). As expected, the genotypes of the offspring mice were $Sun5^{+/-}$ (*Figure 4B*). Similar results were achieved for WT mice (*Figure 4C–D*). The offspring of the $Sun5^{-/-}$ mice and WT mice showed no significant differences in body mass and testis weight (*Figure 4—figure supplement 1A–B*). This strategy might also be suitable for patients with *SUN5* mutation-associated acephalic spermatozoa syndrome, and therefore, we selected the sperm heads rather than those with motile flagella and performed ICSI for two patients with *SUN5* mutations (*Figure 4E and G*) (*Porcu et al., 2003*; *Emery et al., 2004*; *Gambera et al., 2010*; *Saïas-Magnan et al., 1999*). Two women became pregnant, and two healthy babies were born (*Figure 4E and G*), the babies were confirmed to be heterozygous for the *SUN5* mutation (*Figure 4F and H*). These results suggest that *Sun5* mutation-associated infertility could be successfully resolved by ICSI.

## Discussion

In humans, the acephalic spermatozoa syndrome has been reported for decades, and it is characterized by semen that mostly contains sperm flagella without heads; a subtype of this syndrome has been reported in some infertile man with predominantly decapitated or acephalic spermatozoa (*Perotti and Gioria, 1981*; *Perotti et al., 1981*; *Baccetti et al., 1984*; *Chemes et al., 1987*; *Chemes and Alvarez Sedo, 2012*). They are sometimes wrongly denominated

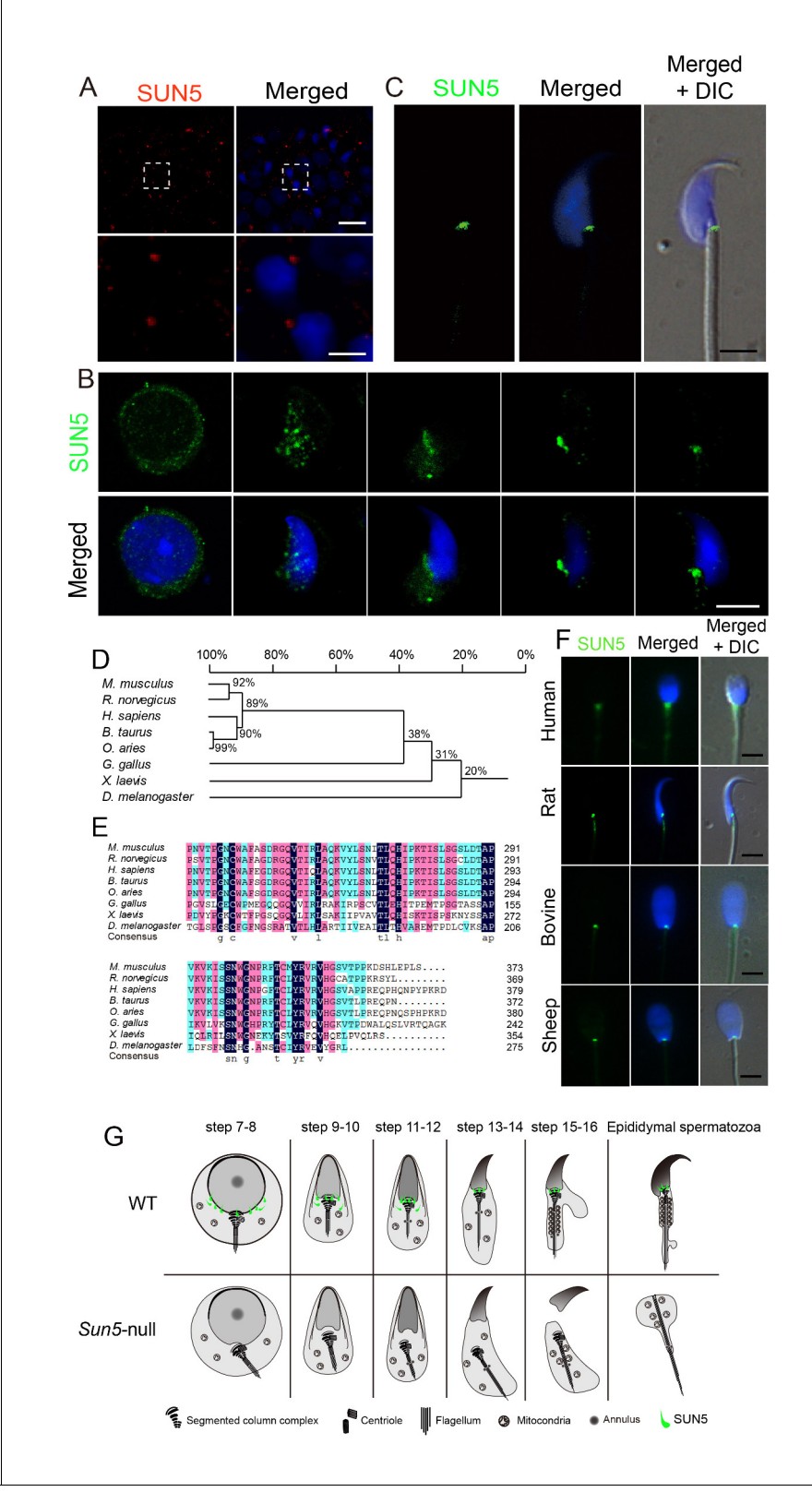

**Figure 3.** SUN5 localizes to the coupling apparatus between the sperm head and tail in mammals. (**A**) IF of SUN5 in testes. Scale bar: upper panel, 10 μm; lower panel, 2.5 μm. (**B**) IF of SUN5 in spermatids at different developmental stages. Scale bar: 5 μm. (**C**) Single-sperm immunofluorescence of SUN5. Scale bar: 5 μm. (**D**) Phylogenetic tree of the SUN5 homolog proteins from different species. (**E**) Sequence alignment of the conserved SUN domain of SUN5 in different species. The dark blue labeled sequences showed 100% identity among species, pink labeled ones showed lower identity

*Figure 3 continued on next page*

*Figure 3 continued*
than the dark blue ones, then the green labeled ones showed lower identity than the pink ones. (**F**) SUN5 localizes to the sperm head-tail coupling apparatus in all tested mammals. Scale bar: 5 μm. (**G**) Schematic representation of the role of SUN5 in the development of the coupling apparatus in WT and *Sun5*-null spermatids based on TEM analyses and immuno-staining.
DOI: https://doi.org/10.7554/eLife.28199.010

globozoospermatozoa; they are actually isolated, headless tails with globular drops of residual cytoplasm, and the etiology of this syndrome subtype is far from complete. Our results showed that the ablation of SUN5 leads to globozoospermatozoa-like acephalic spermatozoa in the mouse model. Together with our previous report and other's paper about *SUN5*-associated mutations in acephalic spermatozoa syndrome patients (*Zhu et al., 2016*; *Elkhatib et al., 2017*), our studies demonstrate that defects in SUN5 may be the major cause of the acephalic spermatozoa syndrome.

Several acephalic spermatozoa-related genes, such as *Odf1* (*Yang et al., 2012*), *Hook1*(named by the hook like phenotype) (*Mendoza-Lujambio et al., 2002*), and *Oaz3* (ornithine decarboxylase antizyme 3) (*Tokuhiro et al., 2009*), have been reported in animal models. However, spermatozoa from the abovementioned gene knockout models are mostly fragile, and none of them uniformly yield 100% acephalic spermatozoa, suggesting that the head-tail coupling apparatus in these spermatozoa is still able to be formed, although it is unstable. *Spata6* (spermatogenesis associated 6) knockout mice were the first mouse model to produce nearly 100% acephalic spermatozoa (*Yuan et al., 2015*). *Spata6* is exclusively expressed in testes and is localized to the coupling apparatus. In the absence of SPATA6, the head-tail coupling apparatus is poorly assembled, and all of the sperm heads and tails are separated during the late stage of spermiogenesis. SUN5 is distinct from all of the above genes, as SUN5 may be localized in the inner membrane of the sperm nuclear envelope, and while the head-tail coupling apparatus can be well-assembled without SUN5, it cannot attach to the nuclear envelope. Therefore, SUN5 might be the inner-most element for flagellum anchoring; in other words, SUN5 is the root of the whole flagellum.

Sperm heads are rarely found in the *Sun5*$^{-/-}$ epididymis because most heads are retained in the seminiferous epithelium when they are separated from the flagellum during spermiation. In addition, due to the loss of efficient anchoring in the last step of spermiogenesis, the last portion of the sperm cytoplasm is not able to be timely removed, which is why all *Sun5*-null spermatozoa carry a portion of cytoplasm at the top of the flagellum. Without normal anchoring and stable orientation, we found that the mitochondrial sheath fails to be properly arranged, and therefore, the axoneme assembly is also affected. A typical cytoplasmic droplet (CD) can be found in the normal ejaculated spermatozoa, while when cytoplasm around the sperm midpiece is present in large amounts, it may impair the sperm function (*Rengan et al., 2012*). From the TEM analysis we can see that the retained cytoplasm droplet in *Sun5*-null spermatozoa is large and contains materials other than mitochondria, which may further impair sperm motility, so the retained cytoplasm droplet may not only be a result of failed spermiogenesis, but also be a second reason for infertility. All the *Sun5* mutated mouse and human spermatozoa are less motile than WT ones, structurally we think it is caused by the misarranged mitochondria inside the mutated spermatozoa, and as mentioned above, the large amount of cytoplasm might also be an obstacle for sperm motility.

Most of the spermatozoa in our recently reported infertile patients with *SUN5* mutations are 'pin headed,' and thus, they are different from the mice model. This difference might come from a partial loss of SUN5 function in those patients. There are two nonsense, one frameshift, two splice-site and five missense mutations in the infertile patients, most of which are homologous mutations or compound heterozygous mutations. The effect of these mutations might be slightly weaker than the frame-shift 13 bp deletion in the mice. The phenotype of the affected individuals was acephalic spermatozoa with a variable but low proportion of abnormal head-tail junctions and tailless heads. Some of the sperm heads in *SUN5*-mutated human sperm samples are linked to the tail but exhibit abnormal structures; these sperm heads do not have implantation fossa and have lost the linear alignment of the sperm axis. Therefore, depletion of *Sun5* leads to the loss of the key element in the sperm head-tail junction, the implantation fossa and basal plate, which is consistent with the phenotype observed in our mouse model.

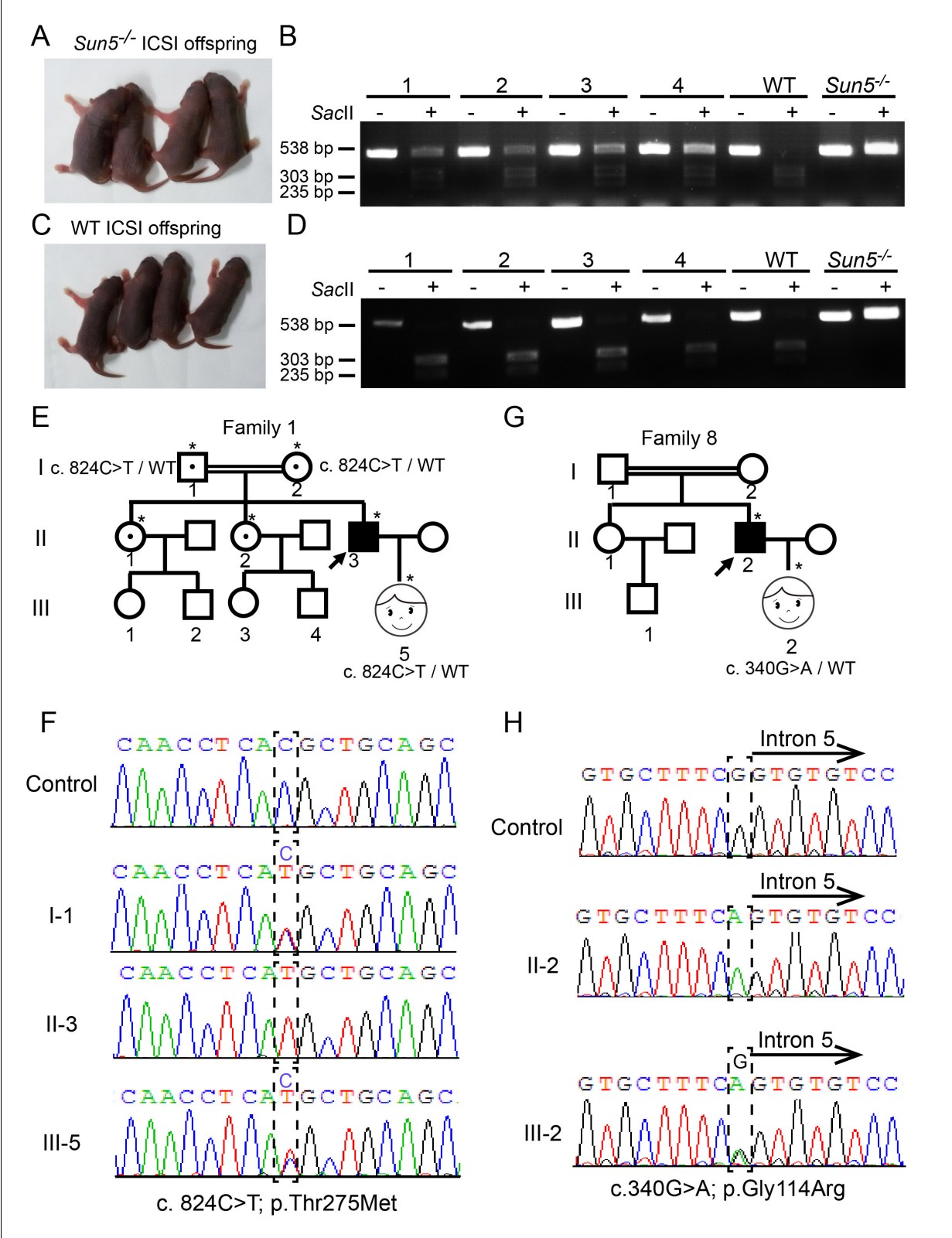

**Figure 4.** Infertility caused by SUN5 mutations could be overcome by ICSI. (**A**) Representative images and (**B**) genotypes of the *Sun5*-null ICSI offspring. (**C**) Representative images and (**D**) genotypes of the WT ICSI offspring. (**E**) Pedigree of family 1 with inherited *SUN5* mutations, and the healthy baby of the infertility patient after ICSI. The individuals with a single star were Sanger sequenced. (**F**) Sequences of the *SUN5* mutation sites of the representative individuals from each generation of family 1. (**G**) Pedigree of family 8 with inherited *SUN5* mutations, and the health baby of the infertility
*Figure 4 continued on next page*

*Figure 4 continued*

patient after ICSI. The individuals with a single star were Sanger sequenced. (**H**) Sequences of the *SUN5* mutation sites of the representative individuals from each generation of family 8.

DOI: https://doi.org/10.7554/eLife.28199.011

The following source data and figure supplement are available for figure 4:

**Source data 1.** The sperm motility and morphology analysis of the two patients underwent ICSI.

DOI: https://doi.org/10.7554/eLife.28199.013

**Figure supplement 1.** Development of WT and *Sun5*-null ICSI offspring.

DOI: https://doi.org/10.7554/eLife.28199.012

Due to the conserved SUN domain, SUN5 was expected to function in meiosis, as is the case with SUN1 and SUN2, and its unclear localization in previous reports led to increased confusion regarding the role of SUN5. SUN1 and SUN2 proteins are relatively larger than the SUN3-SUN5 proteins (*Razafsky and Hodzic, 2009*), and the former play roles in both mitosis and meiosis (*Ding et al., 2007*; *Crisp et al., 2006*; *Padmakumar et al., 2005*) while the latter are restricted to haploid cells (*Göb et al., 2010*; *Calvi et al., 2015*; *Frohnert et al., 2011*). SUN1 and SUN2 directly mediate the movement of chromosomes and the migration of the whole nuclei, while SUN3-SUN5 mainly function in nuclear modeling and integrity. It has been hypothesized that the relatively larger SUN1 and SUN2 might be replaced by the smaller SUN3-SUN5 after meiosis, because with the condensation of the sperm nucleus, the space between the outer and inner layers of the nuclear envelope must be decreased, and therefore, it might not be suitable for the larger SUN1 and SUN2 proteins but is suitable for SUN3-SUN5 (*Sosa et al., 2013*). Our investigations shed light on the *bona fide* function of SUN5 during spermatogenesis. We found that knockout of *Sun5* has no effects on mouse meiosis, acrosome biogenesis or sperm nuclear remodeling, but it specifically destroyed the integrity of the spermatozoa, ultimately resulting in acephalic spermatozoa.

Our current investigation, together with the evaluation of *SUN5*-mutated patients, revealed that SUN5-deficient acephalic spermatozoa syndrome is an autosomal-recessive syndrome, and this type of patient could achieve healthy offspring by ICSI. For the treatment of this type of acephalic spermatozoa syndrome, our studies raised at least two very important issues. First, some *Sun5*-null spermatozoa look very similar to globozoospermatozoa, but there is no chromatin in the top of the flagellum. Since this type of sperms with headless flagella are still motile, they are easily regarded as globozoospermatozoa. To attract more attention, we propose to call this type of spermatozoa pseudo-globozoospermatozoa. In addition, globozoospermia-like sperms need to be carefully checked to see whether there are nuclei, as only those with real heads are able to fertilize oocytes. Second, once a male patient has been diagnosed as having *SUN5* mutations, *SUN5* mutation testing will need to be performed on a genomic sample from his wife before undergoing ICSI to avoid recessive homozygous mutations in their offspring. Our results, together with the previous report about *SUN5* mutations, reveal that SUN5 is essential for the integration of the sperm head to the tail, confirming that SUN5 is one of the main causes of the acephalic spermatozoa syndrome, and most importantly, we successfully found a therapeutic strategy with which to overcome infertility in the affected individuals.

## Materials and methods

### Patients and clinical samples

Consent authorisation for publication has been obtained from the two couples involved in the research. All the research on human subjects have got ethical approval given by Biomedical Research Ethics Committee of Anhui Medical University (Reference number: 20140183). The patients (Family 1:II3 and Family 8:II2) were referred to us for semen analysis after 6 and 10 years of sexual intercourse without conception, respectively. Analyses of more than three semen samples, obtained by masturbation after 3 days of sexual abstinence, showed severe teratozoozpermia. Papanicolaou staining and transmission electron micrographs revealed acephalic spermatozoa (or decapitated tails) with a variable but low proportion of intact spermatozoa with an abnormal head-tail junction. Normal-shaped acrosomes were found on the sperm heads. A DNA fragmentation assay using the

flow cytometric sperm chromatin structure assay (SCSA) revealed normally condensed chromatin (*Larson et al., 2000*).

### Patient 1 (Family 1:II3)
The patient and his wife were 28 and 27 years old, respectively. They had been unable to conceive over a period of 6 years. Both had a normal phenotype, no history of significant illness and a normal karyotype. The wife had regular menses, normal hysterosalpingography and a normal hormonal assessment. The man is from a family of three children, and his father and mother are first cousins. Both of his sisters have two children without fertility problems.

### Patient 2 (Family 8:II2)
The patient and his wife were 34 and 35 years old, respectively. They had been unable to conceive over a period of 10 years. Both had a normal phenotype, no history of significant illness and a normal karyotype. The spouse had regular menses, normal hysterosalpingography and a normal hormonal assessment. The man is from a family of two children, and his father and mother are first cousins. His sister has one child without fertility problems.

## The generation of *Sun5* knockout mice
Production of Cas9 mRNA and sgRNA was as performed as previously described (*Shen et al., 2013*; *Chang et al., 2013*). The T7 promoter and the guiding sequence were added to the sgRNA by PCR amplification using the following primers: SUN5 For: 5'TAATACGACTCACTATAGGTCACC TGGCCGCGGTCACGTTTTAGAGCTAGAAATAGC3' and Tracr rev: 5'AAAAAAAGCACCGACTCGG TGCCAC3'. B6D2F1 (C57BL/6 X DBA2, RRID:IMSR_JAX:100006) female mice and ICR mice were used as embryo donors and foster mothers, respectively. Superovulated female B6D2F1 mice (6–8 weeks old) were mated with B6D2F1 stud males, and the fertilized embryos were collected from the oviducts. Cas9 mRNA (100 ng/μl) and sgRNA (20 ng/μl) were injected into the cytoplasm of fertilized eggs with well-recognized pronuclei in M2 medium (Sigma, M7167-50ml, Santa Clara, CA). The injected zygotes were cultured in KSOM (modified simplex-optimized medium, Millipore) with amino acids at 37°C under 5% $CO_2$ in air, and then, 15–25 blastocysts were transferred into the uterus of pseudopregnant ICR females. To genotype the newborns, a 538 bp fragment harboring the PAM sequence and a *Sac*II site was amplified from the genome and digested by the *Sac*II enzyme. The WT genome digests into two fragments (303 bp and 235 bp) while the mutated genome remains undigested, as illustrated in Supplemental Figure 1. The genotyping primers were as follows: forward: 5'CAAGTCTAGGACTCGGGGTGACAGTG3' and reverse: 5'CCTAACTAGGTCACATCACCC-CAGC3'. All of the animal experiments were performed according to approved institutional animal care and use committee (IACUC) protocols (#08–133) of the Institute of Zoology, Chinese Academy of Sciences.

## Antibodies
The rabbit anti-SUN5 polyclonal antibody (17495–1-AP, RRID:AB_1939754) was purchased from Proteintech (Rosemont, IL). The mouse anti-GAPDH antibody (ab1019t)was purchased from Boaoruijing (Beijing, China).The mouse anti-sp56 antibody (55101, RRID:AB_130101) was purchased from QED Bioscience (San Diego, CA). The Afaf antibody was as acquired as previously described (*Wang et al., 2014*).

## Fertility
Fertility was tested in the male mice of the different genotypes (8–12 weeks, n = 6). Each male mouse was caged with two wild-type CD1 females (4–6 weeks), and vaginal plug was checked every morning. Once a vaginal plug was identified (day 1 postcoitus), the male was allowed to rest for 2 days, after which another female was placed in the cage for another round of mating. The plugged female was separated and single caged, and the pregnancy results were recorded. If a female did not generate any pups by day 22 postcoitus, it was deemed as not pregnant and euthanized to confirm that result. The fertility test lasted for 3 weeks.

## Epididymal sperm count and sperm motility assays

The caudal epididymis was dissected from adult mice. Sperms were squeezed out from the caudal epididymis and incubated for 30 min at 37°C in 5% $CO_2$. The incubated sperm medium was then diluted 1:500 and transferred to a hemocytometer for counting. The sperm motility assay was performed as previously described (*Shang et al., 2016*); unfixed sperms were spread onto precoated slides for morphological observation.

Semen sample analysis of human subjects were performed as described (*Tang et al., 2017*), Semen volume and sperm concentration and motility were evaluated according to the World Health Organization (WHO) guidelines. The percentages of morphologically normal and abnormal spermatozoa were evaluated according to the WHO guidelines.

## Transmission electron microscopy.

The transmission electron microscopy samples were prepared as previously described (*Shang et al., 2016*). Ultrathin sections were cut on an ultramicrotome, stained with uranyl acetate and lead citrate, and observed using a JEM-1400 transmission electron microscope (JEOL, Tokyo, Japan).

## Immunofluorescence (IF) and immunohistochemistry (IHC)

The immunofluorescence and immunohistochemical assays were performed as previously described (*Liu et al., 2016*). The IF images were taken immediately using an LSM 780/710 microscope (Zeiss, Oberkochen, Germany) or SP8 microscope (Leica, Wetzlar, Germany). The IHC images were acquired using a Nikon 80*i* inverted microscope equipped with a CCD camera (Nikon, Tokyo, Japan).

## Immunoblotting

Immunoblotting was performed as previously described (*Shang et al., 2016*). The protein lysates (25 mg) were separated by SDS-PAGE and electrotransferred onto a nitrocellulose membrane. The membrane was blocked in 5% skim milk (BD, 232100) and then incubated with corresponding primary antibodies and detected by Alexa Fluor 680 or 800-conjugated goat anti-mouse or Alexa Fluor 680 or 800-conjugated goat anti-rabbit secondary antibodies. Finally, they were scanned using the ODYSSEY Sa Infrared Imaging System (LI-COR Biosciences, Lincoln, NE, RRID:SCR_014579).

## Testis smear

The indicated mice (8-week-old) were euthanized by cervical dislocation. The testes were surgically removed and the tunica albuginea was removed from the testes. Then, the testes were digested with 1 mg/ml collagenase and 1 mg/ml hyaluronidase. Cells were dissociated by gentle pipetting, filtered through a 70 µm filter and then pelleted by centrifugation at 500 x *g* for 10 min. Cells were suspended in 1 ml of phosphate-buffered saline (PBS; Gibco, C14190500BT) and fixed with 4% paraformaldehyde (PFA) solution, washed with PBS and, finally, spread onto polylysine-coated slides for staining.

## Periodic acid-schiff (PAS) staining

PAS staining was performed as previously described (*Lu et al., 2010*). Briefly, testes were fixed by perfusing mice with Bouin's fixatives (Polysciences, Warrington, PA). Paraffin sections (5 µm) were cut and then stained with periodic acid-Schiff (PAS) and hematoxylin. Stages of seminiferous epithelium cycle and spermatid development were determined as previously described (*Hess and Renato de Franca, 2008*).

## ICSI

Eight- to twelve-week-old CD1 and B6D2F1 mice (C57BL/6 × DBA/2) were used to prepare mature oocyte donors. Spermatozoa were released from the caudal epididymis using HTF (human tubal fluid) medium. WT spermatozoa were decapitated by mild sonication. WT and SUN5-null sperm heads were collected by centrifugation in 70% Percoll (Sigma, P4937, Santa Clara, CA) followed by three washes in M2 medium. Single sperm heads were picked up from the sperm suspension and injected into WT oocytes using a micromanipulator with a Piezoelectric actuating pipette at RT. Injected oocytes were transferred to the KSOM medium under mineral oil and cultured at 37°C in a

humidified atmosphere with 5% $CO_2$. The injected oocytes were analyzed 5–8 hr after ICSI and transferred into the oviducts of pseudo-pregnant CD1 females that had been mated during the previous night with vasectomized males. Full-term pups derived from ICSI embryos were obtained through natural labor.

### ICSI for SUN5 mutation-associated infertile patients

Written informed consent was provided by the couples who decided to undergo intracytoplasmic sperm injection (ICSI) at our reproductive medicine center. After pituitary desensitization with Triptorelin (Decapepthl 0.05 mg/d, 14d, Ferring Pharmaceuticals, Switzerland), the patients' wives were stimulated using follicle stimulating hormone (FSH) (Puregon, N.V. Organon, The Netherlands). Estradiol plasma levels and follicle growth were monitored every two days, and human chorionic gonadotrophin (HCG, Livzon Pharmaceutical, China) was administered when three or more follicles reached 18 mm in diameter. Oocyte retrieval was performed 36 hr after HCG injection. Sperms were prepared by discontinuous density gradient centrifugation, and the resulting suspension was diluted in 10 µl drops of polyvinyl pyrolidine (PVP) covered with oil.

For patient I, 18 oocytes were retrieved, and there were 17 mature oocytes (MII). After ICSI, we obtained four day 6 blastocysts (4BB, 4BB, 4BB, and 3BB) according to the scoring system of Gardner and Schoolcraft (*Gardner and Schoolcraft, 1999*), and all embryos were frozen. After 5 months, two of the embryos were thawed and transferred in one artificial cycle, using estradiol valerate. Clinical pregnancy was confirmed by the ultrasonographic evidence of a gestational sac with a fetal heartbeat at the seventh week, which led to the birth of a healthy boy, whose birth weight at full-term was 3200 g.

For patient II, eight oocytes were retrieved 36 hr after HCG injection, and all oocytes were at the MII stage. After ICSI, we obtained two day 6 blastocysts (4AB and 3BB) according to the scoring system of Gardner and Schoolcraft (*Gardner and Schoolcraft, 1999*), one embryo (4AB) was fresh transferred, and the other was cryopreserved. An ongoing pregnancy occurred, leading to the birth of a healthy boy, whose birth weight at full-term was 3400 g.

### Statistical analysis

Statistical analyses were conducted using GraphPad PRISM version 5.01 (GraphPad Software, Inc. RRID:SCR_002798). All data were presented as the means ± SEM. The statistical significance of the differences between the mean values for the different genotypes was measured by Student's t-test with a paired, 2-tailed distribution. The data were considered significant when the P value was less than 0.05 (*), 0.01 (**) or 0.001(***).

## Acknowledgements

We thank Dr. Qi Chen and Kui Liu for their critical reading of the manuscript. We thank Tie Yang and Guopeng Wang from core facilities of state key laboratory of membrane biology in the Institute of zoology, for their contributions in TEM analysis. This work was supported by the National Nature Science of China (Grant No. 31471277 and 91649202) and National key R and D program of China (Grant No. 2016YFA0500901).

## Additional information

### Funding

| Funder | Grant reference number | Author |
| --- | --- | --- |
| National Natural Science Foundation of China | 31471277 | Wei Li |
| National Natural Science Foundation of China | 91649202 | Wei Li |
| National Key Research and Development Program | 2016YFA0500901 | Wei Li |

The funders had no role in study design, data collection and interpretation, or the decision to submit the work for publication.

## Author contributions

Yongliang Shang, Resources, Data curation, Software, Formal analysis, Investigation, Visualization, Methodology, Writing—original draft, Project administration, Writing—review and editing; Fuxi Zhu, Resources, Data curation, Software, Formal analysis, Investigation, Methodology, Writing—review and editing; Lina Wang, Data curation, Formal analysis, Investigation; Ying-Chun Ouyang, Ming-Zhe Dong, Data curation, Investigation, Methodology; Chao Liu, Methodology; Haichao Zhao, Software, Formal analysis; Xiuhong Cui, Formal analysis, Investigation, Methodology; Dongyuan Ma, Xiaoyu Yang, Formal analysis, Methodology; Zhiguo Zhang, Investigation, Methodology; Yueshuai Guo, Software, Formal analysis, Methodology; Feng Liu, Conceptualization; Li Yuan, Resources, Validation, Investigation, Methodology; Fei Gao, Conceptualization, Methodology; Xuejiang Guo, Software, Formal analysis, Investigation, Methodology; Qing-Yuan Sun, Conceptualization, Methodology, Writing—review and editing; Yunxia Cao, Resources, Data curation, Supervision, Investigation, Methodology, Project administration, Writing—review and editing; Wei Li, Conceptualization, Resources, Formal analysis, Supervision, Funding acquisition, Investigation, Methodology, Writing—original draft, Project administration, Writing—review and editing

## Author ORCIDs

Feng Liu [ID] http://orcid.org/0000-0003-3228-0943
Wei Li [ID] https://orcid.org/0000-0002-6235-0749

## Ethics

Human subjects: Consent authorisation for publication has been obtained from the two couples involved in the research.Written informed consent was provided by the couples who decided to undergo intracytoplasmic sperm injection (ICSI) at our reproductive medicine center. All the research on human subject has got ethical approval given by Biomedical Research Ethics Committee of Anhui Medical University (Reference number: 20140183)

Animal experimentation: All of the animal experiments were performed according to approved institutional animal care and use committee (IACUC) protocols (#08-133) of the Institute of Zoology, Chinese Academy of Sciences. All surgery was performed under sodium pentobarbital anesthesia, and every effort was made to minimize suffering.

## Decision letter and Author response

Decision letter https://doi.org/10.7554/eLife.28199.015
Author response https://doi.org/10.7554/eLife.28199.016

# Additional files

## Supplementary files

• Transparent reporting form
DOI: https://doi.org/10.7554/eLife.28199.014

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
