## [Decision Letter]

Thank you for submitting your article "Essential role for SUN5 in anchoring sperm head to the tail" for consideration by *eLife*. Your article has been reviewed by three peer reviewers, and the evaluation has been overseen by Fiona Watt as the Senior Editor and Reviewing Editor. The following individuals involved in review of your submission have agreed to reveal their identity: Aminata Toure (Reviewer #1); John J.M. Bergeron (Reviewer #2); Katerina Dvorakova-Hortova (Reviewer #3).

The reviewers have discussed the reviews with one another and the Reviewing Editor has drafted this decision to help you prepare a revised submission.

Summary:

In this manuscript the authors describe the sperm phenotype of *Sun5^-/-^* mutant mice and the localization of SUN5 in sperm cells during spermiogenesis. They also identify the co-chaperone DNAJB13, as an interactor of SUN5 protein. The authors provide a novel and exhaustive dataset on SUN5 protein and its function. They demonstrate that SUN5 is necessary for anchoring the sperm head to the flagellum, and that the absence of SUN5 in the mouse causes male infertility due to acephalic-spermatozoa. The authors also confirm in mouse and in humans that the use of the detached sperm head is successful in ICSI treatment of the male infertility. The manuscript is well written and the experiments are clearly described.

Essential revisions:

1) Several publications have previously characterized DNAJB13 localization and spatio-temporal distribution at sperm annulus during spermiogenesis. DNAJB13 is also well-known to co-localize with Septin proteins at the annulus. The authors state that DNAJB13 localises to the coupling apparatus; however the immunofluorescence labelling seems to indicate staining at annulus. To establish convincingly that DNAJB13 does not locate to the annulus the authors could perform co-immunodetection experiments of SUN5 with one of the septin proteins known to locate at the annulus (Sept1, 4, 6, 7 or 12) in wild type sperm.

2) In the legend to Figure 4, the authors claim that "The SUN5-DNAJB13 interaction is responsible for sperm head-tail integration". However, the authors do not provide sufficient data supporting this hypothesis. AP MS should be done on n=3 separate biological experiments with appropriate quantification of the proteins (ion currents, spectral (peptide) counts). DNAJB13 appears to be lacking in Supplementary file 1 and the Y2H data is too preliminary. Moreover, the spatio-temporal distribution of DNAJB13 is not much altered in *Sun5^-/-^* sperm cells during spermiogenesis. The authors should check the distribution pattern of septin protein in *Sun5^-/-^* mutant sperm (Sept1, 4, 6, 7 or 12).

3) The authors need to improve the sperm motility analysis. It is not clear whether mouse and/or human SUN5 null sperm are motile. CASA data should be provided from both KO mice and human SUN5 deficient sperm, and motility should be discussed.

4) The authors should provide statistical data showing the percentage of tails with sperm heads versus tails only, in both cauda as well as caput and corpus epididymis for mouse sperm and human ejaculate, to deliver bigger picture of phenotype and feasibility of the ICSI approach.

5) The reviewers were not convinced that SUN5-null head shape is totally normal. In Figure 2 the SUN5 null sperm heads appear smaller length-wise and/or differ in hook and post acrosomal shape. They seem more like immature sperm. Morphometric analysis is required to resolve this point. Moreover, the Figure 2—figure supplement 2 images should be larger and all the abbreviations used in panel A should be explained in the legend. Specific immunofluorescence markers should be used to complement the histology. In addition, the orientation of the late spermatids should be discussed: is it due to a lack of possible interaction of SUN5 and DNAJB13 or is it a secondary effect due to the altered position in the seminiferous tubule?

6) Regarding the statement “Because the antibodies against SUN5 and DNAJB13 were both generated from rabbits, we could not directly test their co-localization by immunofluorescence”, there are other commercial antibodies on the market, and the colocalization analysis including Pearson coefficient using super-resolution should be carried out. Co-immunoprecipitation experiments should be performed in order to support the conclusions.

7) Please discuss this paper as it is relevant: Elkhatib RA et al., Homozygous deletion of SUN5 in three men with decapitated spermatozoa. Hum Mol Genet. 2017 May 25. doi: 10.1093/hmg/ddx200).

[Editors' note: further revisions were requested prior to acceptance, as described below.]

Thank you for resubmitting your work entitled "Essential role for SUN5 in anchoring sperm head to the tail" for further consideration at *eLife*. Your revised article has been favourably evaluated by Fiona Watt as the Senior Editor and Reviewing Editor, and two reviewers.

The reviewers all agreed that the manuscript has been improved. However, the new data on DNAJB13 is not compelling, specifically, iBAC of the GST pulldown of SUN5 in wt mice. Although DNAJB13 may be the most abundant testis specific protein, it is not the most abundant protein overall, and making sense of Supplementary file 1 is not straightforward. For example, pyruvate dehydrogenase appears to be the most dominant. In Figure 4, it appears the pull down was only done once. In that case are the iBAC data technical repeats as opposed to different pull downs? The authors should do n=3 pull downs with GST Sun5 and the necessary reciprocal experiment of GST pulldown of DNAJB13 in wt and *Sun5^-/-^* mice (again n=3). The Y2H, mass spectrometry and GST pulldown studies are equivocal and most DNAJB13 does not colocalize with *Sun5*. In view of these concerns, the reviewers request that you delete the section on DNAJB13 from the manuscript.

Further points are as follows. The mass spectrometry data i.e., MaxQuant and iBAC folders (e.g. raw, peaklist, etc.) should be deposited in a public repository, e.g. http://www.proteomexchange.org.

The Figure 2 figure legend is incomplete. Both normal and abnormal heads are revealed in *Sun5^-/-^*mice but not indicated in the legend. The text of the Results indicates a minority were normal heads – what is the proportion?

The legend to Figure 4 needs clarification. Has the indicated separation of DNA JB13 in *Sun5^-/-^*been measured with quantification as compared to wt?

The supplementary tables need more description in the legends, especially the supplementary file with the mass spectrometry results.

---

## [Author Response]

Essential revisions:1) Several publications have previously characterized DNAJB13 localization and spatio-temporal distribution at sperm annulus during spermiogenesis. DNAJB13 is also well-known to co-localize with Septin proteins at the annulus. The authors state that DNAJB13 localises to the coupling apparatus; however the immunofluorescence labelling seems to indicate staining at annulus. To establish convincingly that DNAJB13 does not locate to the annulus the authors could perform co-immunodetection experiments of SUN5 with one of the septin proteins known to locate at the annulus (Sept1, 4, 6, 7 or 12) in wild type sperm.

Thanks for your suggestions. DNAJB13 was found to co-localize with SEPT4 in annulus, but only in developing spermatids, not mature spermatozoa (Guan J, et al. BMC Dev Biol. 2009): DNAJB13 and SEPT4 localize in the neck region at step 9 spermatids, and DNAJB13 signals migrate with annulus moving towards distal position at step 15. This is similar to our findings. To reveal the possible relationship between SUN5 and SEPT4, we performed co-immunofluorescence of them in wild type spermatozoa. We found that SUN5 and SEPT4 were first detected in the neck region of developing spermatids, and Sept4 then migrated along the annulus, but SUN5 stayed still in the neck region. In the late stage spermatids, SEPT4 was detected partially in the annulus, and partially in the neck region next to the SUN5 signal (Figure 4—figure supplement 2B). The localization and migration of SEPT4 in our result were consistent with the previous results, the overlapping of SUN5 signals and SEPT4 signals implies the possible interaction between them, but using yeasts two hybrid we found that there was no direct interaction between SUN5 and SEPT4 (Figure 4—figure supplement 2C). So, our results together with others reveal that SUN5, DNAJB13 and SEPT4 all localize to the neck region in the developing spermatids, then DNAJB13 and SEPT4 migrate to the annulus. So, DNAJB13 might have multiple functions, which might be dependent on its interaction with SUN5 or SEPT4.

2) In the legend to Figure 4, the authors claim that "The SUN5-DNAJB13 interaction is responsible for sperm head-tail integration". However, the authors do not provide sufficient data supporting this hypothesis. AP MS should be done on n=3 separate biological experiments with appropriate quantification of the proteins (ion currents, spectral (peptide) counts). DNAJB13 appears to be lacking in Supplementary file 1 and the Y2H data is too preliminary. Moreover, the spatio-temporal distribution of DNAJB13 is not much altered in Sun5^-/-^ sperm cells during spermiogenesis. The authors should check the distribution pattern of septin protein in Sun5^-/-^ mutant sperm (Sept1, 4, 6, 7 or 12).

a) We realized that it is overstated, so we changed it to “SUN5 interacts with DNAJB13 during spermiogenesis”.

b) As suggested, we have repeated the GST pulldown assay and MS analysis, and the identified proteins were quantified as required. DNAJB13 is included in the Supplementary file 1 and its Protein ID is Q80Y75. The Y2H data further confirmed the MS results, and provided strong evidence for SUN5-DNAJB13 interaction. Consistent with the Y2H data, the interaction between SUN5 and DNAJB13 was further confirmed by immunoprecipitation of endogenous SUN5 by DNAJB13 antibody in testis. We also overexpressed DNAJB13 and SUN5 in Hela cells, DNAJB13 could be partially recruited to the nuclear envelop by SUN5, but DNAJB13 itself could not be recruited to the nuclear envelope (Figure 4—figure supplement 2A).

c) The distribution of DNAJB13 was altered in SUN5-null spermatids, as illustrated in Figure 4 and the manuscript. In WT spermatids, DNAJB13 was rapidly enriched in the coupling apparatus with the elongation of the spermatid, and it was tightly attached to the nucleus during the maturation of the spermatid (Figure 4, top), which is almost exactly the same as the distribution of SUN5. While in Sun5-null spermatids, although DNAJB13 was enriched to the coupling apparatus, its tight association with the nucleus was never observed, and in the late-stage spermatid, DNAJB13 was only found in the headless tail spermatozoa (Figure 4, bottom). In the revised version we marked the gap between nucleus and DNAJB13 in SUN5-null spermatids to make this defect could be easily found.

As required, we also examined the distribution pattern of SEPT4 in SUN5-null spermatids (Figure 4—figure supplement 2B). In SUN5-null spermatids, SEPT4 firstly appeared as a dot inside the spermatid, but different from WT, SEPT4 did not localize to the neck region of SUN5-null spermatids, because the neck region has been departed from the nucleus. Structural details of the sperm head and tail separation can be found in Figure 2. Despite the altered position of SEPT4 in SUN5-null spermatids, the absence of SUN5 does not influence the maturation of annulus and the migration of SEPT4, since SEPT4 can also be found in annulus in late stage spermatids (Figure 4—figure supplement 2B).

3) The authors need to improve the sperm motility analysis. It is not clear whether mouse and/or human SUN5 null sperm are motile. CASA data should be provided from both KO mice and human SUN5 deficient sperm, and motility should be discussed.

The sperm motility of the mouse and human SUN5-null spermatozoa was measured by CASA, which showed that SUN-defective mouse and human spermatozoa were motile, but their motility was lower than that of the WT spermatozoa. In this revised version, we provided details of sperm motility from CASA, the percentage of Rapid/Medium/Slow/Static spermatozoa were all shown, and the main part of WT and SUN5+/- spermatozoa were Rapid spermatozoa, while most of the SUN5-null mouse spermatozoa were Medium or Slow spermatozoa (Figure 1—figure supplement 2). The motility of SUN5 mutant human spermatozoa was also shown in Figure 5—figure supplement 1, human sperm motility was assessed according to the WHO criteria (4^th^ Edition). From these results we can find that over 70% of the SUN5 mutant human spermatozoa belonged to Grade D (which means static spermatozoa). The motility defects caused by SUN5 knockout or mutations were discussed in the Discussion section.

4) The authors should provide statistical data showing the percentage of tails with sperm heads versus tails only, in both cauda as well as caput and corpus epididymis for mouse sperm and human ejaculate, to deliver bigger picture of phenotype and feasibility of the ICSI approach.

Because spermatozoa in cauda epididymis were considered as mature ones which represents the final phase of sperm development, so usually cauda epididymal spermatozoa were examined to reveal any developmental defects. As to the development of acephalic spermatozoa, the sperm head and tail might breakup during any possible steps, so we examined the spermatozoa from different parts of the epididymis. As required, the percentage of tails with sperm heads versus tails only was measured in both cauda as well as caput and corpus epididymis for mouse sperm and human ejaculate (Figure 2—figure supplement 1), we can hardly find any sperm with head in any part of SUN5 mutant mouse epididymis and human ejaculate. These results indicate that 1) the breakup of the SUN5-null sperm head and tail occurred most possibly in the seminiferous tubules rather than the duct of epididymis because spermatozoa with heads were seldom seen in the caput epididymis; 2) ICSI is the only way to treat the SUN5 mutations associated patients because nearly no intact sperm could be found in the ejaculate.

5) The reviewers were not convinced that SUN5-null head shape is totally normal. In Figure 2 the SUN5 null sperm heads appear smaller length-wise and/or differ in hook and post acrosomal shape. They seem more like immature sperm. Morphometric analysis is required to resolve this point. Moreover, the Figure 2—figure supplement 2 images should be larger and all the abbreviations used in panel A should be explained in the legend. Specific immunofluorescence markers should be used to complement the histology. In addition, the orientation of the late spermatids should be discussed: is it due to a lack of possible interaction of SUN5 and DNAJB13 or is it a secondary effect due to the altered position in the seminiferous tubule?

Thanks for your suggestions. We have performed morphometric analysis of the WT and SUN5-null sperm heads as recommended. For the morphometric analysis of the sperm head, the width and length of the sperm heads were measured according to a published method (Fisher HS, et al. Nat Commun. 2016), and we found that SUN5-null sperm heads were somehow different from the WT ones, the SUN5-null sperm heads were shorter but wider than the WT, and the length/width ratio was smaller than the WT ones (Figure 2—figure supplement 1). We also analyzed the sp56 staining pattern in SUN5-null sperm heads, finding that over 80% of sperm heads showed normal sp56 staining indicating the well-formed acrosome, while only less than 20% of the spermatozoa showed impaired sp56 staining with different patterns (Figure 2—figure supplement 1).

Back to the question “whether SUN5-null sperm heads are immature sperms”, from the current data we cannot say yes or no, from the acrosome development (sp56 in single sperm and AFAF staining in testis) the SUN5-null sperm head seemed mature, and the disrupted sp56 distribution and altered length/width ratio might be resulted from the transportation in the duct of epididymis, because tail-less heads are actually dead heads which should be degraded by the Sertoli cells. But one fact should not be ignored, that is the sperm head and tail are separated in the late stage seminiferous tubules before the final maturation of the sperm in epididymis. So we intend not to judge whether the SUN5-null sperm heads are mature or not, but to discuss the specific function of SUN5 in acrosome biogenesis and the whole spermiogenesis.

As suggested, we have adjusted the images in Figure 2—figure supplement 2 and provided explanations for abbreviations in the figure. Images in Figure 2—figure supplement 2 are PAS staining rather than HE staining which could stain the developing acrosome, and acrosome itself could serve as a marker of the spermatid development. As a complement to the PAS staining we also stained the acrosome by FITC labeled PNA (Peanut agglutintin) to show the orientation of the sperm head (Figure 2—figure supplement 2).

Spermatids in the late stage tubules especially in stage VII-VIII were well aligned with their heads and acrosomic system oriented toward the basement membrane, this orientation required the well-assembled sperm head-coupling apparatus-tail. The coupling apparatus was destroyed in the late stage SUN5-null spermatid, the sperm tail was separated from the head, and the head could not align itself as the WT sperm, so we think it’s the loss of the coupling apparatus and sperm tail that leads to the miss-alignment of the sperm head in SUN5-null spermatids.

6) Regarding the statement “Because the antibodies against SUN5 and DNAJB13 were both generated from rabbits, we could not directly test their co-localization by immunofluorescence”, there are other commercial antibodies on the market, and the colocalization analysis including Pearson coefficient using super-resolution should be carried out. Co-immunoprecipitation experiments should be performed in order to support the conclusions.

Thanks for your suggestions, we tried to find other non-rabbit originated commercial antibodies, but failed. One goat poly-antibody against DNAJB13(F-20) was cited in a published paper, but the production has been stopped by the company (Santa Cruz). So we have to immunize mice for 6 weeks using His-FLAG tagged DNAJB13, and the anti-serum against DNAJB13 recovered from mouse blood was tested (Figure 4—figure supplement 1) and then used in the following experiments (Figure 4). The interaction between SUN5 and DNAJB13 was further confirmed by immunoprecipitation of endogenous SUN5 by DNAJB13 antibody in testis (Figure 4). In single developing spermatids, co-localization of SUN5 and DNAJB13 in the sperm head to tail coupling apparatus was observed (Figure 4), and Pearson coefficient is 0.47 ± 0.06.

7) Please discuss this paper as it is relevant: Elkhatib RA et al., Homozygous deletion of SUN5 in three men with decapitated spermatozoa. Hum Mol Genet. 2017 May 25. doi: 10.1093/hmg/ddx200).

Thanks for your reminder, we have cited this recently published paper in Results and Discussion.

[Editors' note: further revisions were requested prior to acceptance, as described below.]

The reviewers all agreed that the manuscript has been improved. However, the new data on DNAJB13 is not compelling, specifically, iBAC of the GST pulldown of SUN5 in wt mice. Although DNAJB13 may be the most abundant testis specific protein, it is not the most abundant protein overall, and making sense of Supplementary file 1 is not straightforward. For example, pyruvate dehydrogenase appears to be the most dominant. In Figure 4, it appears the pull down was only done once. In that case are the iBAC data technical repeats as opposed to different pull downs? The authors should do n=3 pull downs with GST Sun5 and the necessary reciprocal experiment of GST pulldown of DNAJB13 in wt and Sun5^-/-^ mice (again n=3). The Y2H, mass spectrometry and GST pulldown studies are equivocal and most DNAJB13 does not colocalize with Sun5. In view of these concerns, the reviewers request that you delete the section on DNAJB13 from the manuscript.

We have deleted the section on DNAJB13 from the manuscript, and related information has been deleted in figures, legends, and figure supplements. Related descriptions have been corrected.

Further points are as follows. The mass spectrometry data i.e., MaxQuant and iBAC folders (e.g. raw, peaklist, etc.) should be deposited in a public repository, e.g. http://www.proteomexchange.org.

The mass spectrometry proteomics data have been deposited to the ProteomeXchange Consortium (http://www.proteomexchange.org) via the PRIDE partner repository with the dataset identifier PXD007815. Project name: Identification of SUN5 interactors from mouse testis lysate; Project accession: PXD007815.

The Figure 2 figure legend is incomplete. Both normal and abnormal heads are revealed in Sun5^-/-^ mice but not indicated in the legend. The text of the Results indicates a minority were normal heads – what is the proportion?

We have rewritten the Figure 2 figure legend, both normal and abnormal heads were included. The proportion of the normal heads had been provided in Figure 1. In the revised version, this information is also provided in the text relating to Figure 2, and the figure legend.

The legend to Figure 4 needs clarification. Has the indicated separation of DNA JB13 in Sun5^-/-^ been measured with quantification as compared to wt?

In WT spermatids, the DNAJB13 signal was tightly adjacent to the nucleus, it is hard to measure the distance between them; while in the Sun5-null spermatid, DNAJB13 signal is obviously detached from the nucleus, and their distances were dynamic according to developmental stages, so the quantification of these distances may not be necessary.

The supplementary tables need more description in the legends, especially the supplementary file with the mass spectrometry results.

We have supplied the necessary descriptions to Figure 1—source data 1 and Figure 4—source data 1 including the MS results and the sperm motility and morphology analysis of the two patients underwent ICSI.

For MS results:

The list showed the proteins uniquely identified in top and bottom band with statistical differences by intensity based absolute quantification (iBAQ), and their Gene ontology annotation. Noting that DNAJB13 is the most abundant identified testis-specific protein, and the only gene annotated to be related to “sperm flagellum”, “axoneme”, or “motile cilium”. Blank gels at corresponding molecular weight in lane 2 of Figure 4 were used as control in the iBAQ analysis. (However, because this part is related to DNAJB13, we deleted it in this version.)

For sperm motility and morphology analysis of the two patients:

The sperm motility and the percentages of morphologically normal and abnormal spermatozoa were evaluated according to the World Health Organization (WHO) guidelines. Most of the *SUN5*-mutation associated spermatozoa were acephalic sperms with low motility.